# Factors associated with COVID-19 vaccine uptake in people with kidney disease: an OpenSAFELY cohort study

The OpenSAFELY Collaborative, Edward PK Parker,[1] John Tazare,[1] William J Hulme,[2] Christopher Bates,[3] Edward J Carr,[4] Jonathan Cockburn,[3] Helen J Curtis ,[2] Louis Fisher,[2] Amelia CA Green,[2] Sam Harper,[3] Frank Hester,[3] Elsie MF Horne,[5,6] Fiona Loud,[7] Susan Lyon,[8] Viyaasan Mahalingasivam ,[1] Amir Mehrkar,[2] Linda Nab,[2] John Parry,[3] Shalini Santhakumaran,[9] Retha Steenkamp,[9] Jonathan AC Sterne ,[5,6,10] Alex J Walker ,[2] Elizabeth J Williamson,[1] Michelle Willicombe,[11,12] Bang Zheng,[1] Ben Goldacre ,[2] Dorothea Nitsch,[1,9] Laurie A Tomlinson [1]

For numbered affiliations see end of article.

**Correspondence to**
Dr Laurie A Tomlinson;
laurie.tomlinson@lshtm.ac.uk

## ABSTRACT

**Objective** To characterise factors associated with COVID-19 vaccine uptake among people with kidney disease in England.

**Design** Retrospective cohort study using the OpenSAFELY-TPP platform, performed with the approval of NHS England.

**Setting** Individual-level routine clinical data from 24 million people across GPs in England using TPP software. Primary care data were linked directly with COVID-19 vaccine records up to 31 August 2022 and with renal replacement therapy (RRT) status via the UK Renal Registry (UKRR).

**Participants** A cohort of adults with stage 3–5 chronic kidney disease (CKD) or receiving RRT at the start of the COVID-19 vaccine roll-out was identified based on evidence of reduced estimated glomerular filtration rate (eGFR) or inclusion in the UKRR.

**Main outcome measures** Dose-specific vaccine coverage over time was determined from 1 December 2020 to 31 August 2022. Individual-level factors associated with receipt of a 3-dose or 4-dose vaccine series were explored via Cox proportional hazards models.

**Results** 992 205 people with stage 3–5 CKD or receiving RRT were included. Cumulative vaccine coverage as of 31 August 2022 was 97.5%, 97.0% and 93.9% for doses 1, 2 and 3, respectively, and 81.9% for dose 4 among individuals with one or more indications for eligibility. Delayed 3-dose vaccine uptake was associated with younger age, minority ethnicity, social deprivation and severe mental illness—associations that were consistent across CKD severity subgroups, dialysis patients and kidney transplant recipients. Similar associations were observed for 4-dose uptake.

**Conclusion** Although high primary vaccine and booster dose coverage has been achieved among people with kidney disease in England, key disparities in vaccine uptake remain across clinical and demographic groups and 4-dose coverage is suboptimal. Targeted interventions are needed to identify barriers to vaccine uptake among under-vaccinated subgroups identified in the present study.

## STRENGTHS AND LIMITATIONS OF THIS STUDY

⇒ We harnessed unique electronic health data linkages to provide a comprehensive overview of demographic and clinical factors associated with COVID-19 vaccine coverage in a large cohort (n=992 205) of people with moderate-to-severe kidney disease.

⇒ We used a gold-standard registry of people receiving treatment for end-stage kidney disease (the UK Renal Registry) to identify dialysis and transplant recipients at the start of the UK COVID-19 vaccine roll-out.

⇒ We estimated cumulative 3-dose and 4-dose coverage estimates in subgroups defined by ethnicity, social deprivation status, age and kidney disease severity to identify under-vaccinated populations.

⇒ The under-representation of certain regions (such as London) in the OpenSAFELY-TPP database may limit generalisability of our findings.

⇒ Eligibility for a fourth COVID-19 vaccine dose could not be precisely determined based on diagnosis and prescription codes in electronic health records.

## BACKGROUND

Since the emergence of SARS-CoV-2, individuals with kidney disease—in particular those requiring renal replacement therapy (RRT)—have been at particular risk of enduring the most harmful effects of COVID-19.[1 2] While vaccination has the potential to mitigate this risk, people with kidney disease exhibit impaired COVID-19 vaccine immunogenicity and effectiveness compared with healthy adults.[3 4] Among individuals who have received a 2-dose primary vaccine series, chronic kidney disease (CKD) and receipt of RRT have been identified as key risk factors

for subsequent infection, COVID-19-related hospitalisation and COVID-19-related mortality.[5]

Approximately 2.6 million individuals over 16 years of age are estimated to be affected by stage 3–5 CKD in England (prevalence of ~6%).[6] Given their continued risk of COVID-19 after primary vaccination, people with kidney disease have been prioritised for additional vaccine doses. The UK Joint Committee on Vaccination and Immunisation (JCVI) recommended that individuals with substantial immunosuppression (including primary or acquired immunodeficiency and recent or ongoing immunosuppressive therapy)[7] receive a third primary vaccine dose from 1 September 2021. A subset of individuals with kidney disease, such as those receiving immunosuppressive therapy following kidney transplant, would have been eligible for this third primary dose. Subsequently, all individuals with CKD were included in the initial JCVI priority groups to be offered a booster dose from 14 September 2021,[8] with eligibility later extending to all adults as of 29 November 2021.[9]

Within the UK's guidelines, there were two routes to being offered a fourth dose of COVID-19 vaccine before September 2022. First, severely immunosuppressed individuals who received a third primary dose would become eligible for a booster (fourth) dose at an interval of 3 months, and a second booster (fifth) dose at least 3 months later. Second, from February 2022,[10] individuals who were >75 years of age, care home residents or immunosuppressed become eligible for a fourth dose as a 2-dose primary course followed by autumn 2021 and spring 2022 boosters. As CKD incidence climbs with age, the majority of people with kidney disease in the UK became eligible for a fourth dose via this second route.

Global and national variation in COVID-19 vaccine uptake is monitored via several online dashboards.[11–13] In England, relative vaccine uptake among demographic and clinical subgroups, including people with CKD, has been reported in weekly coverage reports using the OpenSAFELY platform.[13 14] However, in these summary reports, it remains unclear whether COVID-19 vaccine coverage varies according to kidney disease severity or in relation to other demographic or clinical factors in this high-risk group.

To address these issues, we conducted a retrospective cohort study within the OpenSAFELY-TPP database (https://opensafely.org). We report on the status of COVID-19 vaccination among people with kidney disease up to 31 August 2022 and highlight individual-level characteristics associated with vaccine uptake.

## METHODS

### Data sources

All data were linked, stored and analysed securely within the OpenSAFELY platform (https://opensafely.org/). OpenSAFELY is a data analytics platform created by our team on behalf of NHS England to address urgent COVID-19 research questions. The dataset analysed within OpenSAFELY-TPP is based on 24 million people currently registered with general practice (GP) surgeries using TPP SystmOne software. Data include pseudonymised data such as coded diagnoses, medications and physiological parameters. No free text data are included. All code is shared openly for review and reuse under MIT open license (https://github.com/opensafely/ckd-coverage-ve). Detailed pseudonymised patient data is potentially reidentifiable and therefore not shared. Primary care data are linked through OpenSAFELY with other pseudonymised datasets, including COVID-19 testing records via the Second Generation Surveillance System, Accident and Emergency attendance and hospital records via NHS Digital's Hospital Episode Statistics, national death registry records from the Office for National Statistics and RRT status via the UK Renal Registry (UKRR). Vaccination status is directly available within GP records via the National Immunisation Management System.

### Study population

Baseline clinical and demographic characteristics were defined as of 1 December 2020—the month in which the UK COVID-19 vaccination programme commenced—with the exception of age, which was calculated as of 31 March 2021 as per UK Health Security Agency recommendations.[15] We assessed eligibility for inclusion in the study cohort among individuals who were at least 16 years of age and had been registered in OpenSAFELY-TPP for at least 3 months at baseline. We excluded individuals if their sex, ethnicity, geographical region or index of multiple deprivation (IMD) were unknown, or if they had an atypical vaccine record (more than five doses recorded by 31 August 2022 or any two doses recorded within a period of 14 days).

We determined kidney function based on the most recent serum creatinine measurement in the 2 years preceding 1 December 2020 (where available) and converted this into estimated glomerular filtration rate (eGFR) using the CKD epidemiology collaboration equation without specification of ethnicity.[16] Individuals receiving RRT were identified based on their treatment modality (dialysis or transplant) within the UKRR as of 31 December 2020 (the UKRR status update closest to the baseline date).

We included individuals who were receiving RRT as well as people with CKD based on evidence of reduced kidney function (eGFR <60 mL/min/1.73 m$^2$) in the absence of RRT. Individuals with primary care codes suggesting prior receipt of dialysis or kidney transplant but absent from the UKRR cohort were excluded. To assess CKD severity, we distinguished between stage 3a (eGFR of 45–59 mL/min/1.73 m$^2$), stage 3b (eGFR of 30–44 mL/min/1.73 m$^2$) and stage 4–5 CKD (eGFR <30 mL/min/1.73 m$^2$).

### Outcomes

We report on cumulative coverage for doses 1, 2, 3, 4 and 5 (via any combination of vaccine products) up to 31 August 2022. For the analysis of individual-level factors

associated with vaccine uptake, our primary outcome was receipt of a 3-dose series of any vaccine product. For the purposes of this study, we do not distinguish between third primary and booster doses, or between 100 μg (full) or 50 μg (half) doses of the vaccine product mRNA-1273 (with the latter recommended for booster doses).

Receipt of a 4-dose series of any vaccine product was considered as a secondary outcome. Notably, not all members of our cohort would have been eligible for a fourth dose before September 2022 based on UK guidelines (described above). We therefore limited our secondary outcome analysis to a subset of the population with at least one indicator for receipt of a fourth dose (3 primary+1 booster or 2 primary+2 boosters), including individuals who were ≥75 years of age at baseline, care home residents, transplant recipients or had a history of haematologic malignancy or immunosuppression.

## Covariates

Covariates of interest in relation to demography, vaccine prioritisation and population accessibility included: age (five categories defined by JCVI priority group cut-offs); sex; ethnicity (white, black, South Asian, mixed or other); social deprivation based on IMD quintile (derived from an individual's postcode at Lower Super Output Area); setting (urban, urban conurbation or rural); care home residence; health and social care worker status; housebound status; and receipt of end-of-life care.

Clinical covariates of interest included kidney disease subgroup (CKD3a, CKD3b, CKD4–5, RRT (dialysis) and RRT (transplant)) and presence or absence of pre-existing conditions including: prior SARS-CoV-2 infection; immunosuppression (defined by any prior immunosuppressive diagnosis or receipt of immunosuppressive therapy in the 6 months preceding baseline); moderate/severe obesity (body mass index ≥35 kg/m$^2$ based on the most recent measurement in the 5 years before baseline); diabetes; chronic respiratory disease (including asthma); chronic heart disease; chronic liver disease; asplenia; non-haematologic cancer; haematologic cancer; non-kidney organ transplant; chronic neurological disease (including learning disability); severe mental illness; and classification as clinically extremely vulnerable[17] in the absence of any of the conditions listed above (including RRT). To assess the potential influence of primary care coding relating to kidney disease, we included the presence or absence of diagnostic codes associated with stage 3–5 CKD at baseline.

## Statistical analysis

We used Kaplan-Meier estimates to determine cumulative dose-specific vaccine coverage over time, censoring at death, deregistration or the end of the study period (31 August 2022). Individuals remained in the at-risk population for receiving doses 2–5 regardless of their vaccination status for prior doses. To identify demographic and clinical covariates associated with time to receipt of a 3-dose (primary outcome) or 4-dose (secondary outcome) vaccine series, we used Cox proportional hazards models, stratified by region and censoring at death, deregistration or the end of the study period. For each covariate, we determined hazard ratios (HRs) and 95% CIs with sequential adjustment for baseline covariates. Minimally adjusted models included age, care home residence and health and social care worker status given their use in JCVI prioritisation criteria. Partially adjusted models extended these models to include additional variables potentially linked with vaccine access or uptake, including housebound status, receipt of end-of-life care, setting, sex, ethnicity, IMD quintile, prior SARS-CoV-2 infection, immunosuppression and haematologic cancer. Fully adjusted models included all covariates. Schoenfeld residual plots were used to assess validity of the proportional hazards assumption. As a precaution against small-number disclosures, we rounded counts to the nearest 5, redacted non-zero rounded counts or non-counts of ≤10 and delayed Kaplan-Meier steps until ≥5 events had occurred.

To determine whether HRs differed according to kidney disease severity, partially adjusted models were explored in the following subgroups: CKD3a, CKD3b, CKD4–5, RRT (dialysis) and RRT (transplant). To assess whether conclusions were robust to Cox model assumptions and representative of factors associated with vaccine coverage at the time of the analysis cut-off (31 August 2022), we performed a sensitivity analysis for the primary outcome in which we used partially adjusted logistic regression models (using the same variables above but with region as an additional covariate) among individuals who remained alive and registered throughout follow-up. Finally, we calculated cumulative 3-dose and 4-dose coverage in subgroups defined by ethnicity, IMD quintile, age and kidney disease severity to identify populations at particular risk of under-vaccination at the end of follow-up. While recognising their heterogeneity, minority ethnic groups are combined in this subgroup analysis due to their low combined prevalence (<10%) within the study population.

## Patient and public involvement

Patients were not directly involved in developing this study. We maintain a publicly available website (https://opensafely.org) through which members of the public or patient groups are invited to contact us regarding this study or the broader OpenSAFELY project.

## RESULTS

### Vaccine coverage and product profile by dose

A total of 992 205 individuals met the criteria for inclusion in the cohort (online supplemental figure 1). Of these, 66.5% had CKD3a, 24.7% had CKD3b, 6.5% had CKD4–5, 1.0% were receiving RRT via dialysis and 1.3% were receiving RRT via kidney transplant. Clinical and demographic characteristics of the overall cohort and kidney disease subgroups are summarised in table 1. In

**Table 1** Baseline characteristics of study population

| Characteristic | All, n (%) | CKD3a, n (%) | CKD3b, n (%) | CKD4–5, n (%) | RRT (dialysis), n (%) | RRT (transplant), n (%) |
|---|---|---|---|---|---|---|
| N | 992 205 (100) | 659 785 (100) | 245 380 (100) | 64 855 (100) | 9455 (100) | 12 735 (100) |
| Age | | | | | | |
| 16–64 | 109 815 (11.1) | 76 390 (11.6) | 13 750 (5.6) | 5990 (9.2) | 4395 (46.5) | 9285 (72.9) |
| 65–69 | 74 740 (7.5) | 57 870 (8.8) | 11 270 (4.6) | 3090 (4.8) | 1095 (11.6) | 1415 (11.1) |
| 70–74 | 143 575 (14.5) | 109 375 (16.6) | 25 860 (10.5) | 6025 (9.3) | 1145 (12.1) | 1170 (9.2) |
| 75–79 | 181 200 (18.3) | 130 805 (19.8) | 39 940 (16.3) | 8620 (13.3) | 1225 (13.0) | 615 (4.8) |
| 80+ | 482 870 (48.7) | 285 345 (43.2) | 154 560 (63.0) | 41 135 (63.4) | 1590 (16.8) | 245 (1.9) |
| Sex | | | | | | |
| Female | 555 365 (56.0) | 373 790 (56.7) | 139 545 (56.9) | 33 635 (51.9) | 3490 (36.9) | 4910 (38.6) |
| Male | 436 840 (44.0) | 285 995 (43.3) | 105 835 (43.1) | 31 220 (48.1) | 5965 (63.1) | 7825 (61.4) |
| Ethnicity | | | | | | |
| White | 930 565 (93.8) | 621 555 (94.2) | 231 735 (94.4) | 59 610 (91.9) | 7305 (77.3) | 10 355 (81.3) |
| Black | 17 675 (1.8) | 11 895 (1.8) | 3440 (1.4) | 1270 (2.0) | 575 (6.1) | 490 (3.8) |
| South Asian | 32 620 (3.3) | 19 015 (2.9) | 7720 (3.1) | 3115 (4.8) | 1280 (13.5) | 1490 (11.7) |
| Mixed | 4755 (0.5) | 3195 (0.5) | 975 (0.4) | 340 (0.5) | 120 (1.3) | 125 (1.0) |
| Other | 6590 (0.7) | 4125 (0.6) | 1505 (0.6) | 515 (0.8) | 170 (1.8) | 275 (2.2) |
| Index of multiple deprivation quintile | | | | | | |
| 1—most deprived | 164 415 (16.6) | 104 895 (15.9) | 42 025 (17.1) | 12 265 (18.9) | 2625 (27.8) | 2605 (20.5) |
| 2 | 187 240 (18.9) | 122 190 (18.5) | 47 225 (19.2) | 12 990 (20.0) | 2185 (23.1) | 2650 (20.8) |
| 3 | 224 590 (22.6) | 149 385 (22.6) | 55 965 (22.8) | 14 600 (22.5) | 1890 (20.0) | 2750 (21.6) |
| 4 | 216 980 (21.9) | 146 780 (22.2) | 52 795 (21.5) | 13 355 (20.6) | 1530 (16.2) | 2515 (19.7) |
| 5—least deprived | 198 985 (20.1) | 136 535 (20.7) | 47 370 (19.3) | 11 640 (17.9) | 1225 (13.0) | 2210 (17.4) |
| Setting | | | | | | |
| Urban city or town | 530 975 (53.5) | 353 115 (53.5) | 131 920 (53.8) | 34 580 (53.3) | 4750 (50.2) | 6605 (51.9) |
| Urban conurbation | 204 530 (20.6) | 132 680 (20.1) | 50 665 (20.6) | 14 715 (22.7) | 2995 (31.7) | 3470 (27.2) |
| Rural | 256 700 (25.9) | 173 990 (26.4) | 62 790 (25.6) | 15 560 (24.0) | 1705 (18.0) | 2655 (20.8) |
| Primary care coding of kidney disease | | | | | | |
| CKD3–5 diagnostic code | 592 520 (59.7) | 326 090 (49.4) | 191 500 (78.0) | 57 250 (88.3) | 8030 (84.9) | 9650 (75.8) |
| Dialysis code | 15 785 (1.6) | (N) | (N) | (N) | 7485 (79.2) | 8300 (65.2) |
| Kidney transplant code | 13 630 (1.4) | (N) | (N) | (N) | 1290 (13.6) | 12 340 (96.9) |
| Risk group (occupation/access) | | | | | | |
| Care home resident | 44 470 (4.5) | 24 280 (3.7) | 15 255 (6.2) | 4705 (7.3) | 165 (1.7) | 65 (0.5) |
| Health/social care worker | 5680 (0.6) | 4395 (0.7) | 630 (0.3) | 210 (0.3) | 70 (0.7) | 370 (2.9) |
| Housebound | 54 600 (5.5) | 27 320 (4.1) | 19 140 (7.8) | 6930 (10.7) | 600 (6.3) | 610 (4.8) |
| End of life care | 40 400 (4.1) | 20 555 (3.1) | 13 545 (5.5) | 5515 (8.5) | 605 (6.4) | 180 (1.4) |
| Risk group (clinical) | | | | | | |
| Prior SARS-CoV-2* | 27 835 (2.8) | 15 570 (2.4) | 7830 (3.2) | 2855 (4.4) | 1045 (11.1) | 535 (4.2) |
| Immunosuppression | 56 985 (5.7) | 33 045 (5.0) | 13 600 (5.5) | 4000 (6.2) | 985 (10.4) | 5355 (42.0) |
| Moderate/severe obesity | 111 630 (11.3) | 73 530 (11.1) | 27 730 (11.3) | 7910 (12.2) | 1315 (13.9) | 1150 (9.0) |
| Diabetes | 275 745 (27.8) | 156 995 (23.8) | 82 940 (33.8) | 28 015 (43.2) | 3950 (41.8) | 3840 (30.2) |
| Chronic respiratory disease (including asthma) | 120 935 (12.2) | 75 960 (11.5) | 33 880 (13.8) | 9085 (14.0) | 1015 (10.7) | 990 (7.8) |
| Chronic heart disease | 438 965 (44.2) | 264 765 (40.1) | 128 885 (52.5) | 36 710 (56.6) | 4755 (50.3) | 3850 (30.2) |

Continued

**Table 1** Continued

| Characteristic | All, n (%) | CKD3a, n (%) | CKD3b, n (%) | CKD4–5, n (%) | RRT (dialysis), n (%) | RRT (transplant), n (%) |
|---|---|---|---|---|---|---|
| Chronic liver disease | 32 215 (3.2) | 20 940 (3.2) | 7850 (3.2) | 2290 (3.5) | 550 (5.8) | 585 (4.6) |
| Asplenia | 8665 (0.9) | 5680 (0.9) | 2090 (0.9) | 585 (0.9) | 140 (1.5) | 170 (1.3) |
| Cancer (non-haematologic) | 168 840 (17.0) | 107 245 (16.3) | 46 265 (18.9) | 13 055 (20.1) | 1315 (13.9) | 960 (7.5) |
| Haematologic cancer | 20 430 (2.1) | 12 180 (1.8) | 5830 (2.4) | 1875 (2.9) | 310 (3.3) | 240 (1.9) |
| Organ transplant (non-kidney)† | 2190 (0.2) | 1205 (0.2) | 690 (0.3) | 220 (0.3) | 75 (0.8) | (N) |
| CND (including learning disability) | 163 135 (16.4) | 99 640 (15.1) | 47 650 (19.4) | 13 215 (20.4) | 1450 (15.3) | 1180 (9.3) |
| Severe mental illness | 12 985 (1.3) | 8375 (1.3) | 3275 (1.3) | 1035 (1.6) | 180 (1.9) | 125 (1.0) |
| Clinically extremely vulnerable (other)‡ | 9065 (0.9) | 5765 (0.9) | 2220 (0.9) | 1075 (1.7) | (N) | (N) |
| Region | | | | | | |
| East of England | 220 630 (22.2) | 147 850 (22.4) | 53 895 (22.0) | 13 835 (21.3) | 2085 (22.1) | 2970 (23.3) |
| Midlands | 227 805 (23.0) | 152 235 (23.1) | 55 885 (22.8) | 14 685 (22.6) | 2280 (24.1) | 2715 (21.3) |
| London | 30 610 (3.1) | 19 080 (2.9) | 7315 (3.0) | 2545 (3.9) | 765 (8.1) | 905 (7.1) |
| North East and Yorkshire | 187 440 (18.9) | 123 370 (18.7) | 47 135 (19.2) | 12 615 (19.5) | 1855 (19.6) | 2465 (19.4) |
| North West | 90 180 (9.1) | 59 455 (9.0) | 22 825 (9.3) | 6050 (9.3) | 685 (7.2) | 1160 (9.1) |
| South East | 58 810 (5.9) | 39 085 (5.9) | 14 580 (5.9) | 3830 (5.9) | 580 (6.1) | 735 (5.8) |
| South West | 176 735 (17.8) | 118 715 (18.0) | 43 745 (17.8) | 11 295 (17.4) | 1205 (12.7) | 1780 (14.0) |

Data are n (%) after rounding to the nearest 5. Primary care codes and risk groups are coded by separate binary variables; percentages under these table subheadings therefore do not sum to 100. Among people with CKD4–5, 59 480 had stage 4 CKD (estimated glomerular filtration rate 15–29 mL/min/1.73 m$^2$) and 5375 had stage 5 CKD (estimated glomerular filtration rate <15 mL/min/1.73 m$^2$.
*Based on prior evidence of a positive SARS-CoV-2 test, COVID-19-related primary care code or COVID-19-related hospitalisation as of 1 December 2020.
†Excludes individuals with kidney transplants based on primary care coding or UK Renal Registry status.
‡Classified as clinically extremely vulnerable in the absence of any of the comorbidities listed above (including RRT).
CKD, chronic kidney disease; CND, chronic neurological disease; (N), covariate absent in all individuals by definition; RRT, renal replacement therapy.

comparison to people with CKD, RRT recipients were markedly younger (with associated declines in several age-associated comorbidities) and were more likely to be male and of minority ethnicity.

Cumulative coverage in people with kidney disease as of 31 August 2022 was 97.5% for dose 1, 97.0% for dose 2 and 93.9% for dose 3 (figure 1). Dose 4 coverage was 61.4% for the cohort as a whole and 81.9% among the

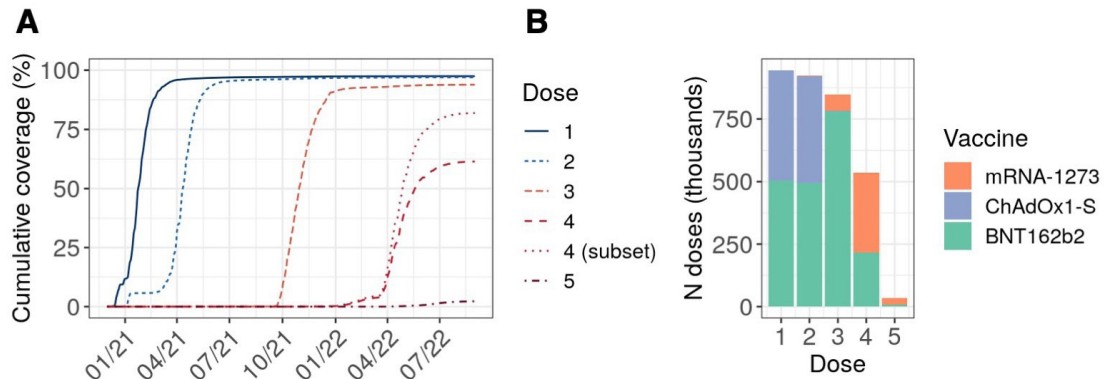

**Figure 1** Dose-specific coverage and product profile among people with kidney disease. (A) Cumulative dose-specific vaccine coverage over time. Cumulative coverage was determined based on Kaplan-Meier estimates for the receipt of any combination of vaccine products, censoring at death, deregistration or 31 August 2022. For dose 4, we report overall coverage (dashed line) and coverage among the subset of the cohort with at least one indicator for receipt of a fourth dose (3 primary+1 booster or 2 primary+2 boosters; dotted line). (B) Vaccine product distribution by dose.

subset of individuals with at least one indicator for receipt of a third primary or second booster dose. Only 19 255 individuals had received a fifth dose, corresponding to a cumulative coverage of 2.3% in the population as a whole.

For their primary vaccine schedule (doses 1 and 2), most individuals received a homologous series of BNT162b2 (53.4% of recorded schedules) or ChAdOx1-S (45.8%), with <1% receiving either heterologous schedules or mRNA-1273 (online supplemental table 1). For subsequent doses, individuals primarily received BNT162b2 (91.8% and 40.4% of recorded doses 3 and 4, respectively) or mRNA-1273 (7.9% and 59.6% of recorded doses 3 and 4, respectively), reflecting JCVI recommendations that RNA vaccines be preferentially used for third primary or booster doses.[7 8] Accordingly, heterologous third doses were prevalent, primarily involving the administration of BNT162b2 or mRNA-1273 to individuals who had received two prior doses of ChAdOx1-S (39.7% and 5.2% of recorded 3-dose schedules, respectively), though homologous vaccination with BNT162b2 remained the most common 3-dose schedule overall (51.7%; online supplemental table 1).

### Factors associated with 3-dose vaccine uptake
Associations between individual-level factors and time to receipt of a 3-dose series are shown in figure 2 and online supplemental table 2. HRs were generally stable across minimally, partially and fully adjusted models (online supplemental table 2). To mitigate potential overadjustment bias, we report on partially adjusted models hereafter.

Vaccine uptake was faster in older individuals and in health and social care workers, reflecting the inclusion of these groups in JCVI prioritisation criteria.[17] By contrast, 3-dose uptake was slower among individuals living in more deprived areas (HR (95% CI) of 1.44 (1.43–1.45) for comparison of least deprived with most deprived IMD quintile), among minority ethnic groups (HRs of 0.48–0.70 relative to white individuals) and individuals who were housebound (HR 0.61 (0.61–0.62)) or receiving end-of-life care (HR 0.87 (0.86–0.88)). These associations were highly consistent across kidney disease subgroups, including RRT recipients (online supplemental figure 2 and online supplemental table 3).

In comparison to people with CKD3a, 3-dose vaccine uptake was somewhat slower among people with CKD3b (HR 0.94 (0.93–0.94)) and CKD4–5 (0.87 (0.87–0.88)), but faster among individuals receiving RRT (HRs of 1.11 (1.08–1.14) and 1.51 (1.48–1.54) for dialysis and transplant recipients, respectively). Having a primary care code associated with stage 3–5 CKD was not strongly associated with vaccine uptake in the cohort as a whole (HR 1.04 (1.04–1.04)), but was associated with faster vaccine uptake in more severe kidney disease subgroups (HRs of <1.1 for CKD3a and CKD3b, 1.17 (1.13–1.20) for CKD4–5, 1.14 (1.07–1.22) for dialysis recipients and 1.11 (1.06–1.16) for transplant recipients; online supplemental figure 2).

Other clinical covariates tended to have modest associations with vaccine uptake (HRs of 0.9–1.1). However, immunosuppression (HR 1.19 (1.18–1.20)) and receipt of non-kidney organ transplants (HR 1.36 (1.30–1.43)) were associated with faster 3-dose uptake, while severe mental illness was associated with slower uptake (HR 0.78 (0.76–0.80)). These associations were highly consistent across CKD and RRT subgroups (online supplemental figure 2).

Cumulative coverage estimates stratified by ethnicity and IMD quintile revealed population subgroups with particularly low vaccination rates (figure 3 and online supplemental table 4). Among minority ethnic groups, 3-dose coverage ranged from 67% in the lowest IMD quintile to 88% in the highest, and was <90% across age and kidney disease subgroups (online supplemental table 4). Among white individuals, 3-dose coverage fell below 90% among individuals <70 years of age and RRT recipients in the most deprived IMD quintiles.

Schoenfeld residual plots revealed no major violations of the proportional hazards assumption. Associations were generally consistent when explored via logistic regression among 838 975 individuals who were uncensored throughout follow-up (online supplemental table 2), with the exception that having a clinical code associated with stage 3–5 CKD was positively correlated with vaccine uptake in the overall cohort using this approach (odds ratio 1.25 (1.23–1.28)).

### Factors associated with 4-dose vaccine uptake
A total of 698 755 individuals (70.4% of the overall cohort) had at least one indicator for receipt of a fourth dose (3 primary+1 booster or 2 primary+2 boosters) and were included in the secondary outcome analysis. Baseline clinical and demographic characteristics for these individuals are summarised in online supplemental table 5.

Factors associated with 4-dose uptake were generally consistent with those observed for the primary outcome analyses. Vaccine uptake was faster in older individuals and in health and social care workers, but slower in individuals living in more deprived areas (HR of 1.52 (1.50–1.53) for comparison of least deprived with most deprived IMD quintile), in minority ethnic groups (HRs of 0.37–0.69 relative to white individuals) and in individuals who were housebound (HR 0.75 (0.74–0.76); online supplemental figure 3 and online supplemental table 6).

Clinical factors associated with eligibility for a third primary dose were also associated with faster 4-dose uptake, including immunosuppression (HR 1.16 (1.14–1.18)), haematologic cancer (HR 1.13 (1.11–1.16)) and transplant (HRs for 2.37 (2.30–2.45) and 1.56 (1.48–1.64) for kidney and non-kidney transplants, respectively). On the other hand, severe mental illness (HR 0.75 (0.72–0.77)) was associated with slower 4-dose uptake. Associations for demographic and clinical covariates were highly consistent across kidney disease subgroups, including RRT recipients (online supplemental figure 4 and online

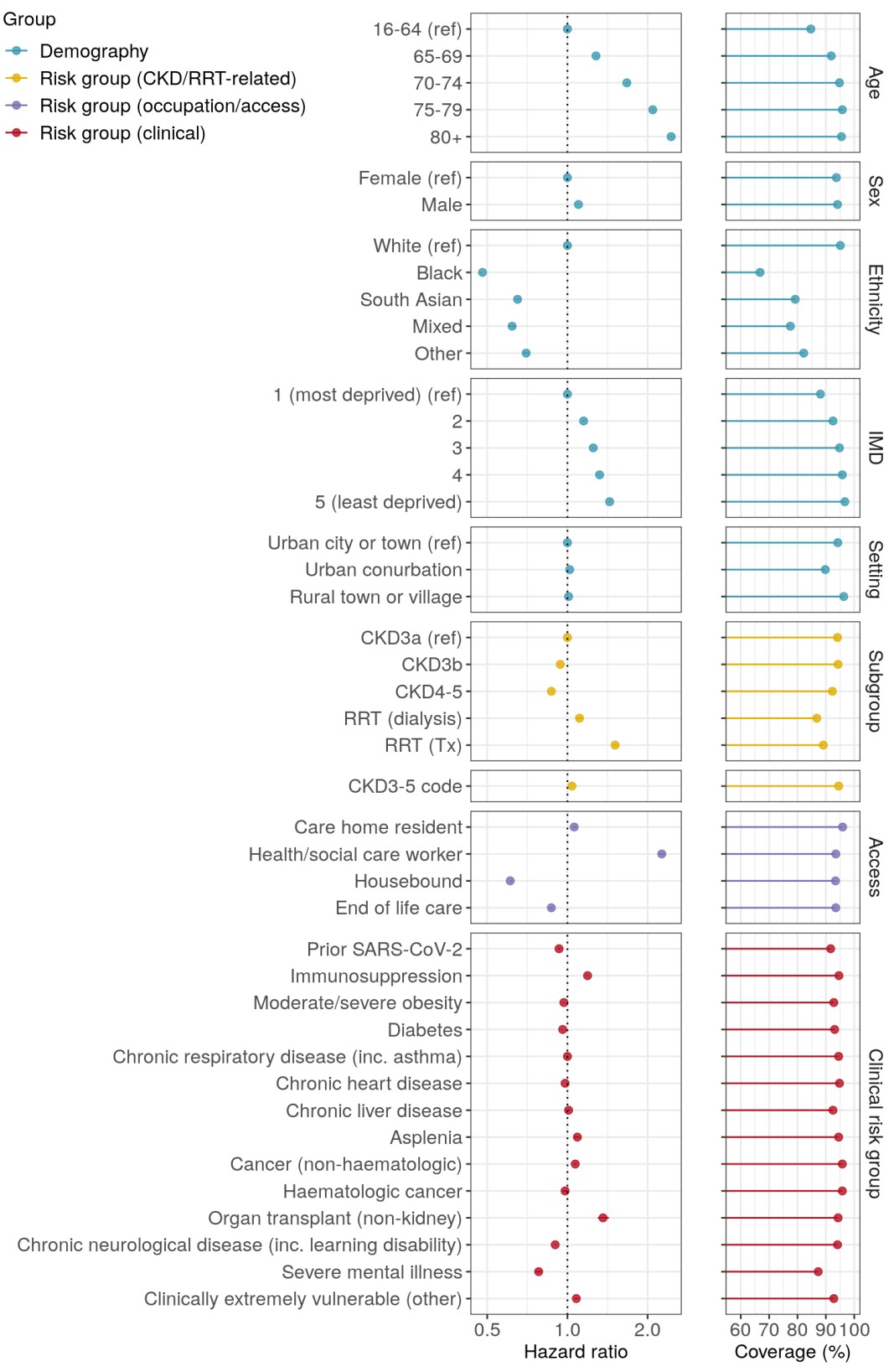

**Figure 2** Factors associated with completion of a 3-dose vaccine series in people with kidney disease. The left-hand panels display hazard ratios and 95% CIs from partially adjusted models that included age, care home residence, health and social care worker status, housebound status, receipt of end-of-life care, setting (urban/rural), sex, ethnicity, IMD quintile, prior SARS-CoV-2 infection, immunosuppression and haematologic cancer. CIs are generally too narrow to be visible. See online supplemental table 2 for minimally and fully adjusted model outputs. The right-hand panels display the cumulative coverage based on Kaplan-Meier estimates, censoring at death, deregistration or 31 August 2022. CKD, chronic kidney disease; IMD, index of multiple deprivation; RRT, renal replacement therapy.

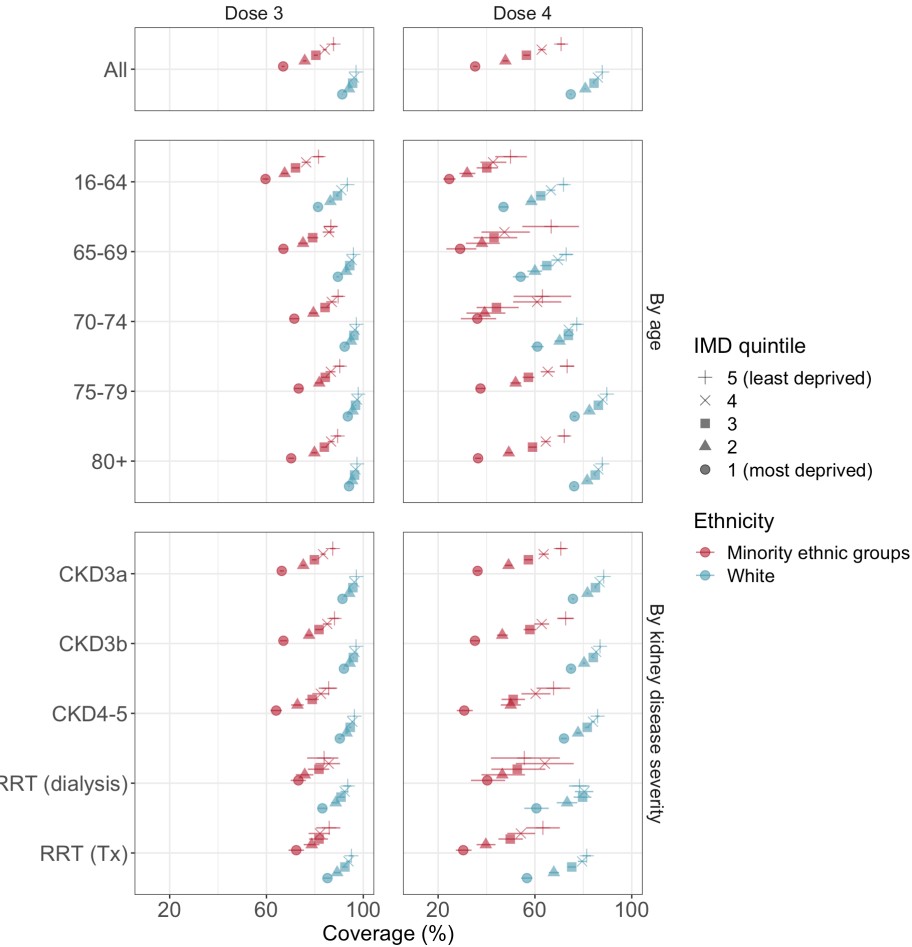

**Figure 3** Cumulative 3-dose and 4-dose coverage in subgroups defined by deprivation index, ethnicity, age and kidney disease severity. Cumulative coverage was determined based on Kaplan-Meier estimates, censoring at death, deregistration or 31 August 2022. Minority ethnic groups are combined due to their low combined prevalence within the study population. See online supplemental tables 4 and 8 for complete data. CKD, chronic kidney disease; IMD, index of multiple deprivation quintile; RRT, renal replacement therapy; Tx, transplant.

supplemental table 7). Having a primary care code associated with stage 3–5 CKD was not associated with 4-dose uptake for the overall cohort, CKD subgroups or dialysis recipients (HRs of 1.03–1.10), but was associated with faster uptake in kidney transplant recipients (HR 1.15 (1.09–1.21)).

Once again, stratified coverage estimates revealed several under-vaccinated subgroups (figure 3). Among minority ethnic groups, 4-dose coverage ranged from 35% in the most deprived IMD quintile to 71% in the least deprived, with particularly low coverage (<50%) among individuals aged 16–64 years in all IMD quintiles (online supplemental table 8). In white individuals, 4-dose coverage was below 75% among individuals <70 years of age in all IMD quintiles, and among RRT recipients in the most deprived IMD quintiles.

## DISCUSSION
### Summary
A principal aim of the COVID-19 vaccine roll-out is to protect those at greatest risk of severe disease. Based on the data presented here, considerable progress has been made towards achieving this goal among people with moderate-to-severe kidney disease in England, with 97.0% of individuals having received at least 2 doses of COVID-19 vaccine, 93.9% having received 3 doses and 81.9% of eligible individuals having received 4 doses as of 31 August 2022.

Although high coverage has been achieved across people with kidney disease, considerable variation in vaccine uptake was evident in relation to the clinical and demographic characteristics of these individuals. Minority ethnicity and social deprivation were associated with delayed vaccine administration and lower final coverage, even in high-risk groups such as dialysis and kidney transplant recipients. Among clinical covariates, uptake was faster in individuals with immunosuppression, but slower in individuals with a history of severe mental illness. These patterns were consistent across stage 3–5 CKD and RRT subgroups, and affected both 3-dose and 4-dose uptake.

## FINDINGS IN CONTEXT

Inequities in COVID-19 vaccine uptake according to ethnicity and social deprivation have previously been reported for the general population,[14] with similar disparities also impacting past national vaccination campaigns in the UK.[18] While our study is among the first to explore variation in COVID-19 vaccine uptake in people with kidney disease, several surveys of haemodialysis patients and kidney transplant recipients in France[19] and the USA[20–22] have reported higher COVID-19 vaccine hesitancy among minority ethnic groups and those living in more deprived areas—findings consistent with the delayed vaccine uptake in these groups reported here.

While prior studies have offered inconsistent findings with respect to the link between mental illness and COVID-19 vaccine uptake,[14 23] these typically focused on 1-dose coverage in the early months of the vaccine roll-out, whereas the current study extends the window of follow-up throughout the primary and booster campaigns and focuses on a high-risk clinical group, limiting comparability.

### Strengths and limitations

By harnessing the OpenSAFELY-TPP database, we incorporated linked health records across a population of 992 205 people with kidney disease, providing a comprehensive and timely view of the clinical and demographics factors associated with COVID-19 vaccine uptake in this high-risk population. We used eGFR levels to define CKD, thereby enabling us to avoid potential bias or imprecision associated with the inconsistent use of diagnostic codes across practices and regions, though the use and reporting of creatinine measurements may also be subject to regional inconsistency. The use of UKRR data also enabled us to audit vaccine uptake among high-risk groups while reducing potential misclassification of current RRT status. Finally, we used complementary statistical modelling strategies to ensure our conclusions were robust to the assumptions of any single approach.

Nonetheless, several limitations of our study should be considered. First, the OpenSAFELY-TPP database does not offer a geographically representative sample of England, with regions such as London (3% of the present cohort) notably under-represented. Thus, region-specific variation in vaccine uptake may not be captured. Second, by considering completion of a 3-dose series as the primary endpoint, our Cox models may contravene the assumption of non-informative censoring, with the receipt of vaccine doses 1 and 2 resulting in a reduced likelihood of COVID-19-related mortality (and associated censoring). However, a sensitivity analysis restricted to uncensored individuals offered a consistent view of the factors associated with vaccine uptake at the end of follow-up. Third, individuals with CKD were identified based on a single serum creatinine measurement. This approach is highly sensitive and minimises delay in identification of affected individuals, but lacks confirmation of chronicity.[24] Finally, pre-existing conditions were considered as either present or absent, thereby failing to capture potentially informative details regarding the timing and severity of these conditions. This may have impacted our ability to accurately determine individuals eligible for a fourth COVID-19 vaccine dose, given that factors such as the timing of immunosuppressive therapy relative to vaccination were included in JCVI eligibility criteria.[7]

### Policy implications and interpretation

Our findings suggest that high-risk clinical risk groups do not escape the entrenched inequities shaping vaccine acceptance and uptake. Targeted activities to promote vaccine uptake among under-immunised communities have increasingly been used to expand the reach of the COVID-19 vaccine roll-out,[25 26] and may yet be crucial towards enhancing booster dose coverage in people with kidney disease. Notably, under-vaccinated populations in the present study included: minority ethnic groups of any age or disease severity in areas of both low and high social deprivation; and white individuals <75 years of age in areas of both medium and high social deprivation. Further research such as qualitative surveys within these under-vaccinated populations could offer a valuable opportunity to identify key barriers to vaccine uptake.

The prevalence of stage 3–5 CKD diagnostic codes varied from 49% in people with CKD3a to 88% in people with CKD4–5. These primary care codes may act as a proxy for GP, NHS or personal awareness of CKD, which is often asymptomatic in its early stages. Indeed, coding status has previously been linked with improved primary care management of CKD, including delivery of influenza and pneumococcal vaccines.[27] Although we did not observe a strong link between CKD3–5 code and vaccine uptake in people with CKD3a or CKD3b, the presence of a diagnostic code was associated with earlier 3-dose uptake in people with CKD4–5 and those receiving RRT. This finding highlights the importance of consistent diagnostic code usage by GPs to promote timely vaccination of high-risk individuals.

Given the reduced immunogenicity of COVID-19 vaccines in people with kidney disease alongside the lower effectiveness of vaccines against the Omicron variant and subvariants,[4 28] three doses may be insufficient to provide long-term protection in this high-risk population. A gradual increase in dose 4 coverage was apparent in early 2022, with a notable uptick following the launch of the spring booster campaign in February 2022.[14] Although not all individuals in the present cohort would have been eligible for a 4-dose series before September 2022, 70% had at least one indicator for receipt of a fourth dose. Cumulative dose-4 coverage in these individuals was 81.9%, and fell short of this level in many subgroups, suggesting that concerted efforts to promote booster dose uptake among people with kidney disease remains essential. Evolving evidence regarding the risk of infection after primary vaccination among people with kidney disease serves as a clear mandate for the necessity of booster doses in these patients.[5] Vaccine-hesitant individuals report personal

concerns over vaccine safety; therefore, clear communication about existing and ongoing research into the safety profile of primary and booster COVID-19 vaccine doses in individuals with kidney disease may help promote vaccine uptake in this population.

## CONCLUSION

This study offers a comprehensive view of the status of the COVID-19 vaccine roll-out in a large cohort of people with kidney disease. To our knowledge, this is the first study to explore the link between individual-level characteristics and COVID-19 vaccine uptake at scale in this high-risk clinical group. Vaccine uptake was significantly slower among minority ethnic groups, in areas of higher social deprivation and in people with severe mental illness. Addressing these disparities during ongoing and future booster campaigns will be essential to protect people with kidney disease from the most harmful sequelae of COVID-19.

**Author affiliations**
[1]London School of Hygiene & Tropical Medicine, London, UK
[2]Bennett Institute for Applied Data Science, Nuffield Department of Primary Care Health Sciences, University of Oxford, Oxford, UK
[3]TPP, Leeds, UK
[4]The Francis Crick Institute, London, UK
[5]Population Health Sciences, University of Bristol, Bristol, UK
[6]NIHR Bristol Biomedical Research Centre, Bristol, UK
[7]Kidney Care UK, Alton, UK
[8]UK Kidney Association, Bristol, UK
[9]UK Renal Registry, Bristol, UK
[10]Health Data Research UK South-West, Bristol, UK
[11]Centre for Inflammatory Disease, Department of Immunology and Inflammation, Imperial College London, London, UK
[12]Imperial College Renal and Transplant Centre, Imperial College Healthcare NHS Trust, London, UK

**Acknowledgements** We are very grateful for all the support received from the TPP Technical Operations team throughout this work, and for generous assistance from the information governance and database teams at NHS England/NHS Transformation Directorate.

**Information governance** NHS England is the data controller for OpenSAFELY-TPP; TPP is the data processor; all study authors using OpenSAFELY have the approval of NHS England. This implementation of OpenSAFELY is hosted within the TPP environment which is accredited to the ISO 27001 information security standard and is NHS IG Toolkit compliant. Patient data has been pseudonymised for analysis and linkage using industry standard cryptographic hashing techniques; all pseudonymised datasets transmitted for linkage onto OpenSAFELY are encrypted; access to the platform is via a virtual private network (VPN) connection, restricted to a small group of researchers; the researchers hold contracts with NHS England and only access the platform to initiate database queries and statistical models; all database activity is logged; only aggregate statistical outputs leave the platform environment following best practice for anonymisation of results such as statistical disclosure control for low cell counts. The OpenSAFELY research platform adheres to the obligations of the UK General Data Protection Regulation (GDPR) and the Data Protection Act 2018. In March 2020, the Secretary of State for Health and Social Care used powers under the UK Health Service (Control of Patient Information) Regulations 2002 (COPI) to require organisations to process confidential patient information for the purposes of protecting public health, providing healthcare services to the public and monitoring and managing the COVID-19 outbreak and incidents of exposure; this sets aside the requirement for patient consent. Taken together, these provide the legal bases to link patient datasets on the OpenSAFELY platform. GP practices, from which the primary care data are obtained, are required to share relevant health information to support the public health response to the pandemic, and have been informed of the OpenSAFELY analytics platform.This project includes data from the UKRR derived from patient-level information collected by the NHS as part of the care and support of kidney patients. We thank all kidney patients and kidney centres involved. The data are collated, maintained, and quality assured by the UKRR, which is part of the UK Kidney Association. Access to the data was facilitated by the UKRR's Data Release Group. UKRR data are used within OpenSAFELY to address a limited number of critical audit and service delivery questions related to the impact of COVID-19 on patients with kidney disease.

**Contributors** Conceptualisation: EPKP, EJC, FL, DN and LAT. Methodology: EPKP, JT, WJH, LF, ACAG, VM, SS, EJW, BG, DN and LAT. Software: WJH, CB, JC, SH, FH, JP and AJW. Formal analysis: EPKP and JT. Data curation: CB, JC, SH, FH, AM, JP, SS, RS, AJW and DN. Writing–original draft: EPKP. Writing–review & editing: EJC, HJC, EMFH, FL, SL, LN, SS, JACS, MW, BZ, DN and LAT. Visualisation: EPKP. Supervision: DN and LAT. Project administration: EPKP and AM. Funding acquisition: EPKP, BG, DN and LAT. LAT and BG are guarantors for this study.

**Funding** This work was jointly funded by the Wellcome Trust (222097/Z/20/Z); MRC (MR/V015757/1, MC_PC-20059, MR/W016729/1, MR/W021420/1); NIHR (NIHR135559, COV-LT2-0073) and Health Data Research UK (HDRUK2021.000, 2021.0157). EPKP received funding from the UKRI COVID-19 Longitudinal Health and Wellbeing National Core Study (Phase 1 LHW-NCS, MC_PC-20059) through a secondment scheme (MR/W021420/1). The views expressed are those of the authors and not necessarily those of the NIHR, NHS England, UK Health Security Agency or the Department of Health and Social Care. Funders had no role in the study design, collection, analysis and interpretation of data; in the writing of the report; and in the decision to submit the article for publication.

**Competing interests** BG's work on better use of data in healthcare more broadly is currently funded in part by: the Bennett Foundation, the Wellcome Trust, NIHR Oxford Biomedical Research Centre, NIHR Applied Research Collaboration Oxford and Thames Valley, the Mohn-Westlake Foundation; all Bennett Institute staff are supported by BG's grants on this work. BG is a Non-Executive Director at NHS Digital. EJW holds grants from MRC.

**Patient and public involvement** Patients and/or the public were not involved in the design, or conduct, or reporting, or dissemination plans of this research.

**Patient consent for publication** Not applicable.

**Ethics approval** This study was approved by the Health Research Authority (REC reference 20/LO/0651) and by the London School of Hygiene & Tropical Medicine's Ethics Board (reference 21863).

**Provenance and peer review** Not commissioned; externally peer reviewed.

**Data availability statement** Detailed pseudonymised patient data is potentially reidentifiable and therefore not shared. Access to the underlying identifiable and potentially reidentifiable pseudonymised electronic health record data is tightly governed by various legislative and regulatory frameworks, and restricted by best practice. The data in OpenSAFELY is drawn from General Practice data across England where TPP is the data processor. TPP developers initiate an automated process to create pseudonymised records in the core OpenSAFELY database, which are copies of key structured data tables in the identifiable records. These pseudonymised records are linked onto key external data resources that have also been pseudonymised via SHA-512 one-way hashing of NHS numbers using a shared salt. Bennett Institute for Applied Data Science developers and PIs holding contracts with NHS England have access to the OpenSAFELY pseudonymised data tables as needed to develop the OpenSAFELY tools. These tools in turn enable researchers with OpenSAFELY data access agreements to write and execute code for data management and data analysis without direct access to the underlying raw pseudonymised patient data, and to review the outputs of this code. All code for the full data management pipeline—from raw data to completed results for this analysis—and for the OpenSAFELY platform as a whole is available for review at github.com/OpenSAFELY. **Code availability** Data management was performed using Python, with analysis carried out using R 4.0.2. Code for data management and analysis, as well as codelists, are archived online (https://github.com/opensafely/ckd-coverage-ve).

**ORCID iDs**
Helen J Curtis http://orcid.org/0000-0003-3429-9576
Viyaasan Mahalingasivam http://orcid.org/0000-0002-3157-1127
Jonathan AC Sterne http://orcid.org/0000-0001-8496-6053
Alex J Walker http://orcid.org/0000-0003-4932-6135
Ben Goldacre http://orcid.org/0000-0002-5127-4728
Laurie A Tomlinson http://orcid.org/0000-0001-8848-9493

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
