## [Reviewer comments · BMJ Open]

ARTICLE DETAILS

TITLE (PROVISIONAL)	Factors associated with COVID-19 vaccine uptake in people with kidney disease: an OpenSAFELY cohort study
AUTHORS	Parker, Edward PK; Tazare, John; Hulme, William J; Bates, Christopher; Carr, Edward J; Cockburn, Jonathan; Curtis, Helen; Fisher, Louis; Green, Amelia CA; Harper, Sam; Hester, Frank; Horne, Elsie MF; Loud, Fiona; Lyon, Susan; Mahalingasivam, Viyaasan; Mehrkar, Amir; Nab, Linda; Parry, John; Santhakumaran, Shalini; Steenkamp, Retha; Sterne, Jonathan; Walker, Alex; Williamson, Elizabeth; Willicombe, Michelle; Zheng, Bang; Goldacre, Ben; Nitsch, Dorothea; Tomlinson, Laurie

VERSION 1 – REVIEW

REVIEWER	Kilada, Samantha University of Liverpool, Institute of Infection, Veterinary, and Ecological Sciences
REVIEW RETURNED	27-Sep-2022

GENERAL COMMENTS	Overall, very well-written and clear information presented. Minor corrections are as follows, For the background: Just curious how much of the general population does this group of people with kidney disease make up? Line 183: commas needed around therefore to make sentence not feel awkward (" ,therefore,") Line 264: reference for JCVI recommendations Line 277: reference needed Line 338-9: reference needed Line 346-7: What does it mean for it to be lower? Why could it be lower? Line 349: What does it mean if it's lower? Line 390: reference needed Line 464: missing closing parenthesis Figures/legends: Why are the legends separated from the figures? Is it possible to have them together? Made for confusing reading.
--

REVIEWER	Webster, Angela The University of Sydney, School of Public Health
REVIEW RETURNED	02-Oct-2022

GENERAL COMMENTS	This is a descriptive analysis of vaccine uptake in people with kidney disease in England, with a focus on equity of vaccination. Somewhat unsurprisingly those disadvantaged in society were also disadvantaged in covid-19 vaccination. Nevertheless it is important to demonstrate, such that solutions may be designed
--

	and tested. Overall, the work is sound in design, statistical analysis, and interpretation, and my comments are therefore minor, and largely relate to presentation and interpretation.  - The description of people of non white race/ethnicity as “non-white” ie in relation to their lack of whiteness. This never sits well with me, and I am sure there must be some editorial guidance on more acceptable descriptors? (This not me making a point about the analysis of white versus other in results, which is largely driven distribution and low numbers of the other groups, which is acceptable) - Intersectional disadvantage – no attempt to measure or estimate? no interaction terms in the model? The analysis tackles standard markers of disadvantage, in a standard way, but doesn’t take the opportunity of framing these in a way that might make the way forward clearer – what are the implications for practice? Can you identify the most likely to be unvaccinated? Then intervention points might be clearer. What about a “non-white” female 70 year old from low SES background relative to a white affluent male 30 year old? I think giving a prediction of the absolute differences might be more impactful. - Measurement/classification of CKD stage. If I have understood correctly this was inferred from one measurement of creatinine only – could the authors reflect on the potential for measurement error here? This should at least be discussed as a limitation - Solutions – I feel the results are not surprising, so in discussion, this could be more of a call for reasonable action with some suggestions. There is data overwhelm here, and I feel the authors could do more to draw the reader to the most pertinent findings, and suggest lines of enquiry/solution-testing strategy in the discussion. The abstract conclusion :identifying how is... is a priority” is a little lame in the context of the massive data they have to draw on. - Tables – T1 needs a formatting make-over. Lose the % signs from every cell and add to column header. T2 – make the model variable clearer? Perhaps spell out what is in each model in footnote as the cross ref of symbols to model is hard for a reader. T3 is too small to interpret. T5 might be better/more informative as a forest plot - Figures – F1 – can you use monochromatic scheme to make the plot lines easier to distinguish? (dots and dashes)? Also add some x-ticks for months. F3 is tiny and impossible to read
--	--

VERSION 1 – AUTHOR RESPONSE

Response to reviewer comments

Reviewer: 1

Miss Samantha Kilada, University of Liverpool

Comments to the Author:

Overall, very well-written and clear information presented. Minor corrections are as follows,

For the background: Just curious how much of the general population does this group of people with kidney disease make up?

Author response: Thank you for the suggestion. We have added the following to the Background section to address this: “Approximately 2.6 million individuals over 16 years of age are estimated to be affected by stage 3–5 CKD in England (prevalence of ~6%)⁶.”

Line 183: commas needed around therefore to make sentence not feel awkward (" ,therefore,")
Author response: We have reviewed the sentence in question and are happy with the current phrasing. We are happy to defer to the editor for additional guidance.

Line 264: reference for JCVI recommendations
Author response: We now cite relevant references (7 and 8) at the end of this sentence.

Line 277: reference needed
Author response: As suggested, we now cite reference 17 (chapter 14 the UK HSA green book), which outlines the JCVI priority groups for primary vaccination in Table 2.

Line 338-9: reference needed
Author response: We have amended the phrasing (replacing "The principal aim..." with "A principal aim...") to better reflect that this reflects a subjective statement rather than a citation of an external source. The final sentence now reads as follows: "A principal aim of the COVID-19 vaccine roll-out is to protect those at greatest risk of severe disease."

Line 346-7: What does it mean for it to be lower? Why could it be lower?

Line 349: What does it mean if it's lower?

Author response: Thank you for these comments. In each case, the conclusions are based on the reported hazard ratios (HRs) in Figure 2 and the Supplementary Tables. The reviewer is correct that the term 'lower' is imprecise – failing to capture the fact that these are inferred from time-to-event analyses. We have therefore replaced 'higher' with 'faster' and 'lower' with 'slower' in this paragraph, reflecting the more precise language used throughout the Results section. We also highlight that the final coverage was lower in ethnic minority groups and in areas of higher social deprivation as follows: "Minority ethnicity and social deprivation were associated with delayed vaccine administration and lower final coverage..."

Line 390: reference needed
Author response: We now cite the relevant reference (7) at the end of this sentence.

Line 464: missing closing parenthesis
Author response: We have added a parenthesis and full stop.

Figures/legends: Why are the legends separated from the figures? Is it possible to have them together? Made for confusing reading.

Author response: We included figure legends in the main text and submitted high-resolution figures separately. We defer to the editor for final placement of figures and accompanying legends during typesetting.

Reviewer: 2
Dr. Angela Webster, The University of Sydney

Comments to the Author:

This is a descriptive analysis of vaccine uptake in people with kidney disease in England, with a focus on equity of vaccination. Somewhat unsurprisingly those disadvantaged in society were also disadvantaged in covid-19 vaccination. Nevertheless it is important to demonstrate, such that solutions may be designed and tested. Overall, the work is sound in design, statistical analysis, and interpretation, and my comments are therefore minor, and largely relate to presentation and interpretation.

- The description of people of non white race/ethnicity as "non-white" ie in relation to their lack of whiteness. This never sits well with me, and I am sure there must be some editorial guidance on more acceptable descriptors? (This not me making a point about the analysis of white versus other in results, which is largely driven distribution and low numbers of the other groups, which is acceptable)

Author response: Thank you for this valuable comment. As indicated by the reviewer, the grouping of non-White individuals in our interpretation was a statistical decision given their low combined prevalence (<10%) in the study population. We have now replaced 'non-White' with 'ethnic minority groups' throughout the manuscript and appendix, as recently recommended in the BMJ (<https://doi.org/10.1136/bmj.m4493>), but are happy to change the phrasing if an alternative expression is preferred.

We have also added the following sentence to the Methods section describing the new intersectional subgroup analysis (see below) to clarify that our decision to combine ethnic minority groups is driven by statistical considerations: "While recognising their heterogeneity, ethnic minority groups are combined in this subgroup analysis due to their low combined prevalence (<10%) within the study population."

- Intersectional disadvantage – no attempt to measure or estimate? no interaction terms in the model? The analysis tackles standard markers of disadvantage, in a standard way, but doesn't take the opportunity of framing these in a way that might make the way forward clearer – what are the implications for practice? Can you identify the most likely to be unvaccinated? Then intervention points might be clearer. What about a "non-white" female 70 year old from low SES background relative to a white affluent male 30 year old? I think giving a prediction of the absolute differences might be more impactful.

Author response: Thank you – this is very valuable feedback. We agree that a more granular analysis of intersectional disadvantage would be a valuable addition. Our view is that an analysis of cumulative coverage in key subgroups is easier to interpret than the inclusion of interaction terms in the statistical models, and draws attention to key under-vaccinated populations that should be the focus of future interventions.

We have therefore added an analysis of 3- and 4-dose coverage in subgroups stratified by ethnicity, IMD quintile, age, and kidney disease severity. This additional analysis is described as follows:

Methods: "Finally, we calculated cumulative 3- and 4-dose coverage in subgroups defined by ethnicity, IMD quintile, age, and kidney disease severity to identify populations at particular risk of under-vaccination at the end of follow-up. While recognising their heterogeneity, ethnic minority groups are combined in this subgroup analysis due to their low combined prevalence (<10%) within the study population."

Results (3-dose coverage): "Cumulative coverage estimates stratified by ethnicity and IMD quintile revealed population subgroups with particularly low vaccination rates (Figure 3 and Supplementary Table 4). Among ethnic minority groups, 3-dose coverage ranged from 67% in the lowest IMD quintile to 88% in the highest, and was <90% across age and kidney disease subgroups (Supplementary Table 4). Among White individuals, 3-dose coverage fell below 90% among individuals <70 years of age and RRT recipients in the most deprived IMD quintiles."

Results (4-dose coverage): "Once again, stratified coverage estimates revealed several under-vaccinated subgroups (Figure 3). Among ethnic minority groups, 4-dose coverage ranged from 35% in the most deprived IMD quintile to 71% in the least deprived, with particularly low coverage (<50%) among individuals aged 16–64 years in all IMD quintiles (Supplementary Table 8). In White individuals, 4-dose coverage was below 75% among individuals <70 years of age in all IMD quintiles, and among RRT recipients in the most deprived IMD quintiles."

Discussion: "Notably, under-vaccinated populations in the present study included: ethnic minority groups of any age or disease severity in areas of both low and high social deprivation; and White individuals <75 years of age in areas of both medium and high social deprivation. Further research such as qualitative surveys within these under-vaccinated populations could offer a valuable opportunity to identify key barriers to vaccine uptake."

Figure 3:

Figure 3. Cumulative 3- and 4-dose coverage in subgroups defined by deprivation index, ethnicity, age, and kidney disease severity. Cumulative coverage was determined based on Kaplan-Meier estimates, censoring at death, deregistration, or 31st August 2022. Ethnic minority groups are combined due to their low combined prevalence within the study population. See Supplementary Tables 4 and 8 for complete data. CKD, chronic kidney disease; dial., dialysis; IMD, index of multiple deprivation quintile; RRT, renal replacement therapy; Tx, transplant.

The raw data underlying Figure 3 are presented in Supplementary Table 4 (3-dose estimates) and 8 (4-dose estimates). We thank the reviewer for raising this issue, and feel that the changes above have substantially strengthened the final manuscript.

- Measurement/classification of CKD stage. If I have understood correctly this was inferred from one measurement of creatinine only – could the authors reflect on the potential for measurement error here? This should at least be discussed as a limitation

Author response: We have added the following limitation to address this issue: “Third, individuals with CKD were identified based on a single serum creatinine measurement. This approach is highly sensitive and minimises delay in identification of affected individuals, but lacks confirmation of chronicity²⁴.”

- Solutions – I feel the results are not surprising, so in discussion, this could be more of a call for reasonable action with some suggestions. There is data overwhelm here, and I feel the authors could do more to draw the reader to the most pertinent findings, and suggest lines of enquiry/solution-testing strategy in the discussion. The abstract conclusion :identifying how is... is a priority” is a little lame in the context of the massive data they have to draw on.

Author response: As detailed in the responses above, we have added additional subgroup analyses and a new Figure 3 to highlight key under-vaccinated subgroups within our study population. These are perhaps the most pertinent findings from a policy perspective, and we

have therefore augmented the Discussion to draw more direct attention to these groups (and the need for specific solution-testing) as follows:

“Notably, under-vaccinated populations in the present study included: ethnic minority groups of any age or disease severity in areas of both low and high social deprivation; and White individuals <75 years of age in areas of both medium and high social deprivation. Further research such as qualitative surveys within these under-vaccinated populations could offer a valuable opportunity to identify key barriers to vaccine uptake.”

We have amended the final sentence of the Abstract to reflect a similar sentiment, as follows: “Targeted interventions are needed to identify barriers to vaccine uptake among under-vaccinated subgroups identified in the present study.”

This comment on data overwhelm is well taken. To address this, we have moved Figures 3 and 4 to the supplementary materials given that they consolidate trends reported elsewhere. We have instead added a new Figure 3 that presents cumulative coverage across key subgroups, as described above.

- Tables – T1 needs a formatting make-over. Lose the % signs from every cell and add to column header. T2 – make the model variable clearer? Perhaps spell out what is in each model in footnote as the cross ref of symbols to model is hard for a reader. T3 is too small to interpret. T5 might be better/more informative as a forest plot

Author response: Thank you for these suggestions. We have made the following changes in line with your suggestions:

- We have removed % signs from all cells in Table 1, Supplementary Table 1, and Supplementary Table 5, clarifying in the column headers that ‘n (%)’ are displayed.
- In Supplementary Tables 2 and 5, we agree that the cross-referencing symbols are unclear. We have therefore removed these and added the following text to the figure legends to interpret: “Minimally adjusted models included age, care home residence, and health and social care worker status given their use in vaccine prioritisation criteria. Partially adjusted models additionally included housebound status, receipt of end-of-life care, setting, sex, ethnicity, IMD quintile, prior SARS-CoV-2 infection, immunosuppression, and haematologic cancer. Fully adjusted models included all covariates”
- We have removed the ‘N (n events)’ columns of Supplementary Table 3 (given their overlap with data presented in Table 1) and increased the font size of the remaining columns to improve readability. We have done the same for Supplementary Table 7.
- Supplementary Figure 4 (formerly Figure 4) plots the partially adjusted model outputs presented in Supplementary Table 6 (formerly Supplementary Table 5). We agree that the visual format is more informative. The supplementary table is intended as a supportive adjunct that also provides event counts alongside outputs from minimally and fully adjusted models.

- Figures – F1 – can you use monochromatic scheme to make the plot lines easier to distinguish? (dots and dashes?)? Also add some x-ticks for months. F3 is tiny and impossible to read

Author response: We have amended Figure 1 as suggested. Figure 3 has been moved to the supplement (as Supplementary Figure 2), and its width expanded to improve readability. Note that underlying data is also presented in Supplementary Table 3.

Editor(s)' Comments to Author:

Please delete the sections about what is known and what the study adds as they are not part of the BMJ Open format.

Author response: As requested, this section has been removed.

Please add a section entitled 'Strengths and limitations of this study' (immediately after the abstract). This section should contain up to five short bullet points, no longer than one sentence each, that relate specifically to the methods. The novelty, aims, results or expected impact of the study should not be summarised here.

Author response: We have added the following section below the abstract:

Strengths and limitations of this study

- We harnessed unique electronic health data linkages to provide a comprehensive overview of demographic and clinical factors associated with COVID-19 vaccine coverage in a large cohort (n = 992,205) of people with moderate-to-severe kidney disease.
- We used a gold-standard registry of people receiving treatment for end-stage kidney disease (the UK Renal Registry) to identify dialysis and transplant recipients at the start of the UK COVID-19 vaccine roll-out.
- We estimated cumulative 3- and 4-dose coverage estimates in subgroups defined by ethnicity, social deprivation status, age, and kidney disease severity to identify under-vaccinated populations.
- The under-representation of certain regions (such as London) in the OpenSAFELY-TPP database may limit generalisability of our findings.
- Eligibility for a fourth COVID-19 vaccine dose could not be precisely determined based on diagnosis and prescription codes in electronic health records.

Other amendments

- (1) To ensure that the data are as contemporary as possible, we have extended the end of follow-up from 11th May 2022 (in the original submission) to 31st August 2022 (the day before the launch of the autumn 2022 booster campaign). This has resulted in minor changes to counts and hazard ratio estimates throughout the paper, though no substantive changes to the final conclusions. All amendments in the main text are tracked. However, to improve readability of the appendix, changes in tables are not tracked here.
- (2) We now report on 7 rather than 9 regions (merging North East and Yorkshire into one region and East and West Midlands into another) to ensure consistency across COVID-19 vaccine analyses in OpenSAFELY (e.g. <http://doi.org/10.1101/2022.11.16.22282396>). This has led to no substantive changes in the results.
- (3) We have amended the tense in several places when referring to the Spring 2022 booster campaign given that the Autumn 2022 booster campaign is now underway.
- (4) We have provided a clearer definition of 'immunosuppression' as follows: "...defined by any prior immunosuppressive diagnosis or receipt of immunosuppressive therapy in the 6 months preceding baseline."

VERSION 2 – REVIEW

REVIEWER	Webster, Angela The University of Sydney, School of Public Health
REVIEW RETURNED	23-Dec-2022
GENERAL COMMENTS	I think the changes you have made in revision have sufficiently addressed my concerns and have greatly improved the manuscript. Its a nice piece of work.